# Thermo-Environmental Performance of Four Different Shapes of Solar Greenhouse Dryer with Free Convection Operating Principle and No Load on Product

**Edwin Villagran** [1],*, **Juan Camilo Henao-Rojas** [2],* and **German Franco** [2]

1 Corporación Colombiana de Investigación Agropecuaria—AGROSAVIA, Centro de Investigación Tibaitata, Mosquera, Cundinamarca 250040, Colombia

2 Corporación Colombiana de Investigación Agropecuaria—AGROSAVIA, Centro de Investigación La Selva, Ríonegro, Llanogrande 054048, Colombia; gfranco@agrosavia.co

* Correspondence: evillagran@agrosavia.co (E.V.); jhenao@agrosavia.co (J.C.H.-R.)

**Abstract:** Solar drying using greenhouse dryers is a viable method from the technical, economic, and environmental perspectives, allowing the drying of agricultural products for conservation purposes in different regions of the world. In Colombia, the drying of aromatic plants such as mint (*Mentha spicata*) is usually done directly and in open fields, which exposes the product to contamination and loss of quality. Therefore, the objective of this research was to use a three-dimensional computational fluid dynamics (CFD-3D) model previously successfully validated and implemented in this work to study the performance of air flow patterns, temperature, and humidity inside four greenhouse-type dryers contemplated for a region with hot and humid climatic conditions. The results found allowed us to observe that the spatial distribution of temperature and relative humidity are related to the air flows generated inside each dryer, therefore, there were differences of up to 7.91 °C and 23.81% for the same evaluated scenario. The study also allowed us to conclude that the CFD methodology is an agile and precise tool that allows us to evaluate prototypes that have not been built to real scale, which allows us to generate useful information for decision-making regarding the best prototype to build under a specific climate condition.

**Keywords:** solar dryer; air flow; temperature; CFD simulation; tropical climate



## 1. Introduction

Solar drying is one of the oldest existing conditioning operations for the processing and conservation of food, species, and medicinal plants. Its use is very popular in many regions worldwide mainly because it requires a low economic investment to carry out the drying process [1,2]. This activity in the last two decades has been even more relevant due to the growth of the world population, which has caused food production systems to become more efficient and reduce post-harvest losses [3].

In aromatic herbs, drying is done as an alternative method of conservation that has as its principle of operation to reduce the moisture content of the leaves after harvest. It is also possible to modify the organoleptic properties such as color, smell, and taste, very relevant characteristics in terms of market, allows to preserve the active principles of the product, and finally reduce the risks of product losses due to the attack of pathogens [4,5].

In developing countries, a common practice is to dry products in yards with free exposition during the hours of sunshine and high radiation. This method has several disadvantages, among which we can mention the dependence of the drying time on the climatic conditions of the region and the exposure of the product to dust, birds, pests, and other external agents that can generate losses, contamination, and even the presence of toxins in the final product [1,6]. One of the drying alternatives that can limit some of these disadvantages mentioned, is drying using greenhouse type dryers, which is also a method that has been used and is popular in many countries of the world [7,8].

Greenhouse solar dryers work by generating heat by capturing the long wavelengths coming from the short wave solar radiation that incidents the dryer's roof structure [4]. Inside the dryer, the product to be dried increases in temperature in the form of sensible heat and then, through the latent heat of vaporization, the humidity present in the product evaporates [9]. These dryers can operate on two operating principles, free convection (passive dryer), and forced convection (active dryer). The passive dryer works from the buoyancy movement generated by the air density changes, air that later leaves the dryer through the ventilation areas arranged on the roof or on the sides. While the active dryer needs the support of one or more fans to extract warm and humid air from inside the structure [4,10,11]. The use of these fans generates a higher degree of microclimatic management inside the dryer, which optimizes drying times and the drying process [4,10,11]. Additionally, this type of active dryers can be complemented with the use of renewable energies and heat accumulators to increase the internal temperature [4,10,11].

In Colombia, the production of plants such as mint (*Mentha spicata*) is mostly directed to the international fresh market and small quantities are dehydrated to provide raw material for the infusion industry or for the manufacture of essential oils. In mint (*Mentha spicata*), the quality of the product is influenced by both the agronomic management before harvest and by other processes carried out in the post-harvest such as drying and grinding of plant material, which are requirements verified by buyers at the commercial level, who usually review relevant aspects such as color, safety, flavor, and aroma [12]. In the case of this species, the chemical components can vary in quantity and quality, depending on the drying method used, since the raw material is subjected to a chemical transformation in its monoterpenoid components [13].

The operation of a greenhouse dryer cannot be generalized as it will depend on the size, geometry, location, local weather conditions, shape of the roof, therefore, its design should be previously analyzed before construction [14]. The behavior of different greenhouse dryer designs can be reviewed in the study developed by Singh, Shrivastava, and Kumar [5]. One of the most robust tools to simulate the microclimatic behavior of agricultural structures not built at full scale is the simulation from computational fluid dynamics (CFD) [15,16]. For example, in the study developed by Noh et al. [15,16] the authors studied the efficiency of three drying configurations in a new prototype industrial-type solar load dryer, finding through numerical simulation that the internal temperatures in the dryer could reach up to 59.8 °C under an intermittent active ventilation configuration, which allowed optimizing the operation of the full-scale dryer.

This methodology has already been used for the thermal study in different studies at a universal level, some specific issues of the CFD models can be reviewed in the study performed by Ramachandran et al. [17], also the use of CFD has been applied to study the dryers of products such as coffee [18], corn [19], fish [20]. In a recent study developed by Vivekanandan et al. [21], the authors evaluated by means of a CFD model the temperature generated inside six greenhouse type dryers, obtaining as a result an adequate adjustment of the simulated temperature with the temperature measured experimentally. The authors also reported that the shape of the dryer directly influences its thermal gain, obtaining as a result a 13% difference in the temperature value between the models that presented the highest and lowest value, respectively.

Therefore, and in accordance with what has already been mentioned, the objective of this work was to analyze the aerodynamic, thermal, and humidity performance inside four greenhouse type dryers in conditions of no product load, to determine the prototype of the dryer that generates the best microclimate conditions for the drying of mint (*Mentha spicata*). The spatial distribution of the study variables was carried out using a CFD-3D simulation model previously successfully validated and that for this research was adapted to the dominant climatic conditions of the study region.

## 2. Materials and Methods

### 2.1. CFD Simulation

This research work implemented a 3D Computational Fluid Dynamics (CFD) model previously validated experimentally and published in the work developed by Villagran et al. [22]. The CFD model was used to determine the thermal, hygrometric, and air flow patterns in four different solar greenhouse dryers, so that based on the results obtained, the most appropriate dryer design to be built at full scale for the climatic conditions evaluated can be identified.

The pre-processing, processing, and post-processing software used was Ansys (ANSYS Inc., Canonsburg, PA, USA), in their respective work interfaces in fluid dynamics. Using for process phase the Ansys fluent, software that discretizes the non-linear partial equations that describe the drying process inside a greenhouse type dryer into linear equations with numerical solution through the finite volume method (VOF).

### 2.2. Description of the Dryer Designs Evaluated

In this study, four different greenhouse dryer designs were selected for evaluation, the detailed geometries of each of the dryers are shown in Figure 1. The dryers evaluated have a covered area of 70 m$^2$ (7 m wide and 10 m long), model 1 (M1) is a dual-roof tunnel-type structure with minimum and maximum heights of 1.5 and 2.6 m, respectively. The external cover will be made of commercial transparent polyethylene for agricultural use, and the internal cover will be made of black polyethylene.

Model 2 (M2) is a tunnel-type structure attached to two span, the minimum and maximum heights of each spans are 1.7 and 2.7 m, model 3 (M3) is a chapel-type structure with a flat roof on two sides, the minimum and maximum heights are 1.5 and 2.5 m; model 4 (M4) is a tunnel-type roof structure with minimum and maximum heights of 1 and 2.7 m, respectively. The dryers M2, M3, and M4 were equipped with a polyethylene cover and lateral ventilation areas on both sides, in the case of M2 and M3 ventilation areas were also contemplated in the roof region, both lateral and roof ventilation areas were covered with an insect-proof porous screen.

In the case of M1, the use of insect-proof porous mesh was contemplated in the internal area, precisely on the intermediate region between the two pillars that support the exterior and interior arches and for the side vents. Likewise, in this model, a roof ventilation area was available in the central zone of the internal roof (Figure 1).

### 2.3. Numerical Model

The flow of a fluid under turbulent regime in steady state inside a dryer operating under a principle of free convection operation is usually modeled by a set of three non-linear partial equations known as Navier–Stokes's equations. These represent conservation of momentum (Equation (1)), mass (Equation (2)) and energy (Equation (3)).

$$\nabla(\rho U) = 0 \tag{1}$$

$$\nabla(\rho U U) = \nabla p + \mu_T \nabla U^2 + \rho g + S_h \tag{2}$$

$$\nabla(-k\nabla \mathrm{T} + \rho C_p T U) = 0 \tag{3}$$

where $T$ is temperature; $g$ is the force of gravity; $U$ is the velocity vector; $k$ is the thermal conductivity; $C_p$ the specific heat; $p$ is the pressure; $\mu_T$ and $\rho$ the dynamic viscosity and density, respectively, and finally $S_h$, which represents the source term, such as a heat transfer effect generated by solar radiation or a linear momentum loss term due to a porous medium.

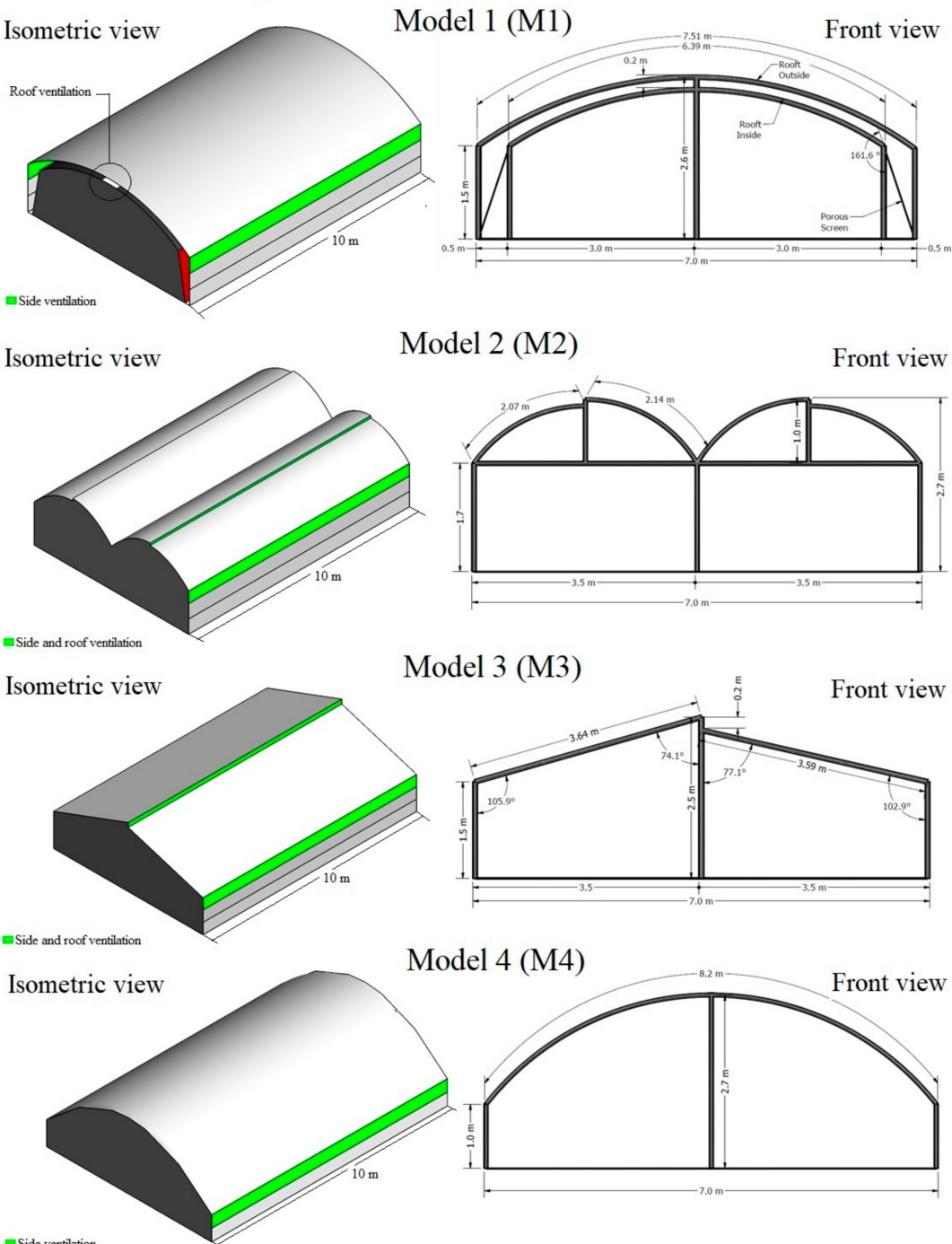

**Figure 1.** Architectural characteristics of the dryers evaluated.

The effect of air buoyancy that allows the description of the free convection phenomenon caused by air density changes is included in the CFD model through the implementation of the Boussinesq approach (Equation (4)).

$$\rho = \rho_0[1 - \beta(T - T_0)] \tag{4}$$

where $\rho$ and $\rho_0$ are the density at a given time and the reference density, respectively; $\beta$ is the coefficient of thermal expansion of the air; $T$ and $T_0$ are the temperature and the temperature of reference. Airflow turbulence was modeled by implementing a Reynolds Averaged Turbulence Model (RANS). For this specific case the standard $k$-$\epsilon$ model was used, which is composed of an equation for dissipation rate (Equation (5)) and another

equation for turbulent kinetic energy (Equation (6)) [23,24]. This model of turbulence has been successfully used in similar works to the one developed in this research [25].

$$\frac{\partial}{\partial t}(\rho \varepsilon) = \frac{\partial}{\partial x_i}\left[\left(\mu + \frac{\mu_t}{\sigma}\right)\frac{\partial \epsilon}{\partial x_i}\right] + \rho C_1 S_\epsilon - \rho C_2 \frac{\epsilon^2}{k + \sqrt{v\epsilon}} + C_{1\epsilon}\frac{\epsilon}{k}C_{3\epsilon}G_b k \tag{5}$$

$$\frac{\partial}{\partial x}(\rho k) = \frac{\partial}{\partial x_j}\left[\left(\mu + \frac{\partial k}{x_j}\right)\frac{\partial k}{\partial x_j}\right] + G_k + G_b - \rho\epsilon - Y_M \tag{6}$$

In these equations, $\mu$ and $\mu_t$ are the viscosity and the turbulent viscosity of the fluid; $G_b$ and $G_k$ represent the generation of turbulent kinetic energy due to buoyancy and velocity in its order. The $\sigma_\epsilon$ and $\sigma_k$ are Prandtl's turbulent numbers for $\varepsilon$ and $k$; $Y_M$ is the fluctuating expansion in turbulence due to the overall dissipation rate, $v$ is the coefficient of kinematic viscosity, $C_1$, $C_2$, $C_{1\epsilon}$, $\sigma$ are constants determined by default in the simulation software [26]. The mass transfer was contemplated by using the species transport model, the conservation equation for the mass balance that defines the humidity conditions is governed by Equation (7).

$$\nabla(\rho Y_i) = -\nabla J + N_i + E_i \tag{7}$$

where $J$ is the diffusion flow of water vapor, $N$ is the rate of water vapor production in the component $i$; $Y$ is the local mass fraction of water vapor in the diffusion-convection equation; and $E$ is the source term for moisture. The porous screens located in the ventilation areas of each dryer were included in the model as porous media. In general, these porous screens are impulse airflow sinks which cause a loss of pressure due to the interaction of the porous material with the fluid flow [27]. This can be modeled using Equations (8) and (9).

$$S_j = -\left(\frac{\mu}{\alpha}\mu_T + C_2\frac{1}{2}\rho|\mu|\mu_T\right) \tag{8}$$

$$C_2 = \frac{k_s}{Y} \tag{9}$$

where $\alpha$ is the permeability of the porous medium; $|\mu|$ is the magnitude of the velocity; $\rho$ is the density of the air; $Y$ is the thickness of the porous medium; $\mu$ is the dynamic viscosity of air; $C_2$ is an inertial resistance factor; $k_s$ is a dynamic parameter that like $C_2$ is dependent on the porosity and architecture of the porous screen material. In Equation (7), with which the equivalent pressure gradient is calculated, the terms $\frac{\mu}{\alpha}\mu_T$ and $C_2\frac{1}{2}\rho|\mu|\mu_T$ are the loss of viscosity and inertial loss, respectively. For this research, the values of $k_s$ and $C_2$ implemented in the numerical model correspond to those used in the porous material modeled in the work developed by [28].

The effect of solar radiation was included in the numerical model through the implementation of the discrete ordinate radiation sub-model (DOM). This radiation model allows solving the radiative transfer equation (RTE) within the computational domain, solving numerically for a finite number of discrete solid angles associated with a vector direction in the three-dimensional Cartesian plane [29]. The RTE is therefore considered a field equation for one wavelength and is defined from Equation (10).

$$\nabla\left(I_\lambda\left(\vec{r}, \vec{s}\right) \quad \vec{s}\right) + (a_\lambda + \sigma_s)I_\lambda\left(\vec{r}, \vec{s}\right)$$
$$= a_\lambda n^2\frac{\sigma T^4}{\pi} + \frac{\sigma_s}{4\pi}\int_0^{4\pi} I_\lambda\left(\vec{r}, \vec{s}\,\prime\right)\Phi\left(\vec{s}\cdot\vec{s}\,\prime\right)d\Omega\prime \tag{10}$$

where $\lambda$ is the radiation at a wavelength; $\sigma$ is Stefan–Boltzmann's constant; $\vec{r}$, $\vec{s}$ are the vectors that indicate the position and direction, respectively; $\vec{s}\,\prime$ is the direction vector of the scatter; $\sigma_s$, $a_\lambda$ are the coefficients of dispersion and spectral absorption of the participating medium. $I_\lambda$ is the spectral intensity of the radiation that depends on $\vec{s}$ and

$\Rightarrow$; $n$ is the index of refraction; $\nabla$ is the divergence operator and finally $\Phi, T,$ and $\Omega$ are the phase function, the local temperature (°C), and the solid angle, respectively.

The semi-implicit method (SIMPLE) for pressure-linked equations was used to solve the coupled pressure-velocity equations, in the case of the discrete gradient the Green–Gauss node-based solution scheme was implemented. The discretization scheme considered for the energy, water vapor, turbulence, drive, and discrete ordinates DO transport equations was a second-order scheme, this in order to ensure more accurate results [30]. For the numerical model, convergence criteria were established for the residuals of $10^{-6}$ for all the evaluated variables, as well as the relaxation factors shown in Table 1. To simplify the numerical model and according to the purpose of this research work, the dryers were simulated in conditions without load of product to be dried (empty dryer).

**Table 1.** Relaxation factors used in the CFD simulation.

| Variable | Relaxation Factors |
|---|---|
| Density | 1.0 |
| Body force | 0.9 |
| Pressure | 0.3 |
| Momentum | 0.7 |
| DO | 0.8 |
| Energy | 0.8 |
| $k$ and $\varepsilon$ | 0.7 |
| Water vapor | 0.7 |
| Turbulent viscosity | 0.8 |

*2.4. Discretization of the Computer Domain and Boundary Conditions*

The numerical grid of the computer domain and its specific conditions of size and quality are an essential step that defines the process of convergence of the numerical solution and the appropriate prediction of the results obtained [31,32]. In this research, the discretization of the four computational domains that included each one of the evaluated dryers, was made from the decomposition of the large computational domains in a finite number of small control volumes integrated in an unstructured grid in a three-dimensional space.

The size of the computational domain was defined from the recommendations given by other authors who used this simulation approach in their research. The size of the computational domain should be defined as a function of the maximum height ($H_{max}$) of the evaluated structure. Therefore, the edges of the computational domain must satisfy the following conditions: the edges parallel to the direction of air flow evaluated must have a minimum dimension of 10 $H_{max}$, the minimum height of the computational domain must have a minimum length equivalent to 10 $H_{max}$. The dimension from the airflow inlet and outlet edge to the evaluated structure must be at least 10 Hmax and 15 Hmax in length, respectively [22,33,34]. According to the above, the dimensions of the computational domain established in this research are shown in Figure 2.

The appropriate size of the numerical grids was defined once the test of independence of the solution to the size and number of elements of each three-dimensional grid evaluated. This procedure must be carried out with the objective of guaranteeing numerical solutions adjusted to the real operation of the structure and to determine the size of the grid that allows to obtain such solutions with the lowest possible computational cost [35,36]. The quality of the four defined numerical grids was evaluated from the orthogonality parameter finding that according to the results obtained in the numerical grids were in the high quality ranges [37], view Table 2.

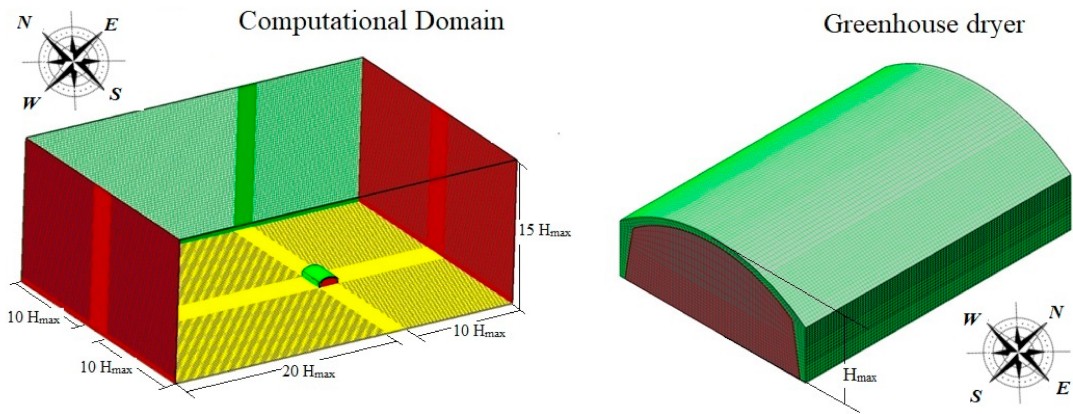

**Figure 2.** Details of the dimensions of the computational domain.

**Table 2.** Size and quality parameters of the numerical grids established in the simulations of each dryer design.

| Model | Number of Elements of the Numerical Grid | Quality of Numerical Grid |
|:---:|:---:|:---:|
| M1 | 3.928.756 | Average: 0.94; maximum: 0.97; minimum: 0.76. |
| M2 | 3.872.906 | Average: 0.93; maximum: 0.98; minimum: 0.74. |
| M3 | 3.956.191 | Average: 0.95; maximum: 0.99; minimum: 0.79. |
| M4 | 3.741.873 | Average: 0.93; maximum: 0.98; minimum: 0.73. |

The edge conditions established in the computational domain were symmetry conditions for the north and south boundaries, a non-slip wall condition was established for the floor of the computational domain. The east side limit was established with an air inlet condition, where a logarithmic wind speed profile was implemented according to what was already defined in the study developed by Villagran, Leon, Rodriguez, and Jaramillo [22]. The west boundary was defined as an area with a pressure outlet and airflow condition, while the roof region was established as a wall condition with a radiation flux established according to the climatic conditions of the study region.

For the structure of the dryer, wall conditions were established for the side, frontal, and cover regions, establishing in the cover a double wall condition which is surrounded by a fluid on both sides and establishing this region as a semitransparent surface for the radiation phenomenon. In the region of the dryer floor, a wall condition with opaque medium behavior was established for the radiation phenomenon. Finally, in the ventilation areas, specific boundary conditions were established for a porous medium. The physical and optical properties of the materials included in the computer domain are listed in Table 3.

## 2.5. Climate Performance of the Study Region and Simulated Scenarios

The region where the dryer will be built is the municipality of Jardin in the southwest of the department of Antioquia—Colombia. Figure 3 shows the main climatic variables for a multiannual period of 30 years. The average value of the annual temperature was 19.2 °C, with average minimum and maximum values of 15.0 °C and 25.4 °C, respectively (Figure 3).

**Table 3.** Thermo-physical and optical properties of materials in the computational domain. Taken from Osorio H et al. [38].

| Variable | Concrete | Black Plastic Cover | Clear Plastic Cover |
|---|---|---|---|
| Density ($\rho$) (kg m$^{-3}$) | 2100 | 920 | 920 |
| Thermal conductivity (k) (W m$^{-1}$ K$^{-1}$) | 1.41 | 0.30 | 0.30 |
| Specific heat (Cp) (J K$^{-1}$ kg$^{-1}$) | 880 | 1800 | 1900 |
| Absorptivity coefficient (1/m) | 0.60 | 9.9 | 0.10 |
| Refractive index | 1.00 | 1.79 | 1.00 |
| Emissivity | 0.71 | 0.95 | 0.90 |
| Scattering coefficient (1/m) | −10 | 0.00 | 0.00 |

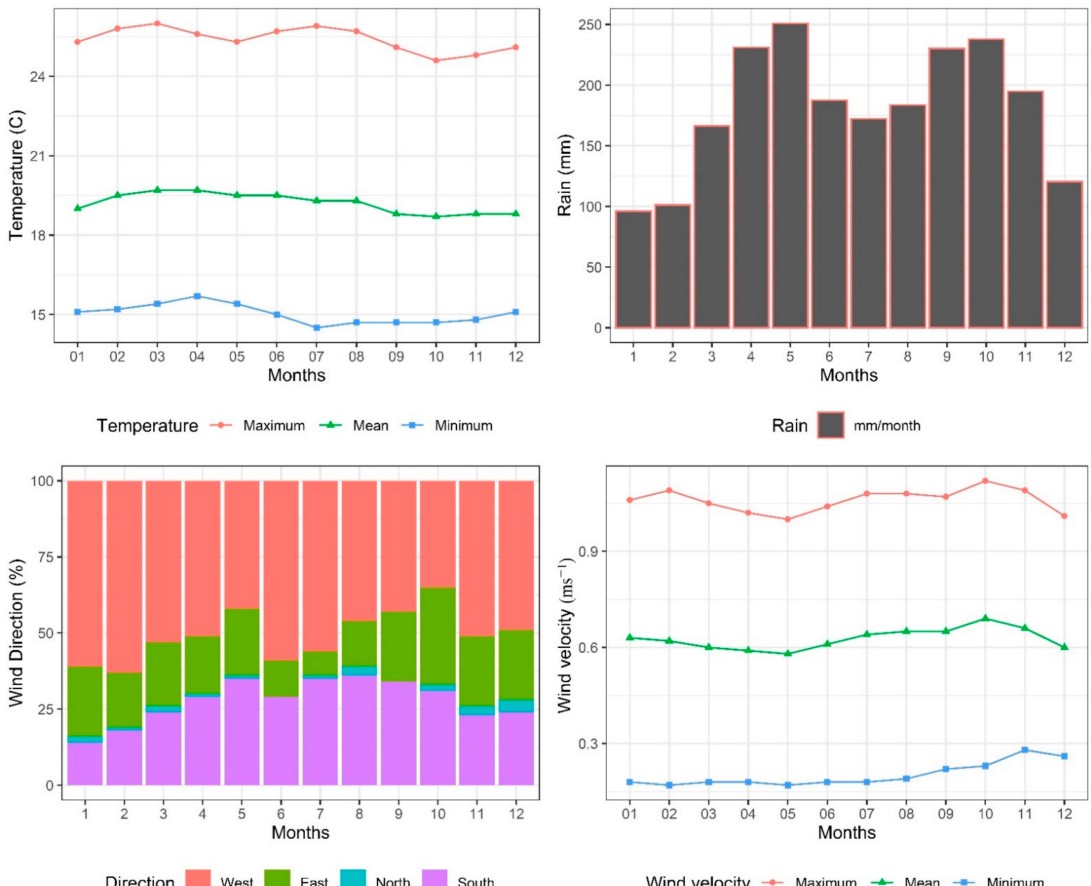

**Figure 3.** Multiannual variation on a monthly scale of the main climate variables in the study region.

The accumulated annual precipitation rate was 2172.4 mm, the precipitation is distributed in a bimodal way with two defined tops over the months of April–May and September–October, where the value of the accumulated monthly precipitation for each of these months is higher than 200 mm, therefore, the average relative humidity presents values higher than 75%. According to the Koppen–Geiger climate classification and the observed temperature and precipitation data, the study region presents a tropical equatorial (*Af*) climate condition characterized by warm and humid conditions [39]. The wind speed shows an average velocity of 0.62 ms$^{-1}$, with minimum and maximum average values of 0.21 ms$^{-1}$ and 1.05 ms$^{-1}$, respectively (Figure 3). Likewise, the dominant wind direction is west (W), with monthly event frequency values above 45% of the total events presented (Figure 3).

According to the mentioned climatic conditions, it was determined that to study the performance of the four dryers, the values obtained for the average maximum temperature, the average relative humidity value, and the three wind speed values indicated in the

most dominant flow direction would be used as initial simulation conditions. On the other hand, not having a radiation value, it was defined that this would be chosen through the solar calculator available in the simulation software. For this, a specific day of the month of July was chosen, which is the highest temperature for 12 o'clock in the day, which is the moment of greatest intensity of solar radiation in the region of study. In this way, the simulation scenarios summarized in Table 4 were formed.

**Table 4.** Initial conditions used in the CFD simulation of each of the scenarios contemplated.

| Scenario | Temperature (°C) | Relative Humidity (%) | Wind Speed (ms$^{-1}$) | Wind Direction | Solar Radiation (wm$^{-2}$) |
|---|---|---|---|---|---|
| M1S1 M1S2 M1S3 | 25.4 | 75 | 0.21 0.62 1.05 | W | 893 |
| M2S1 M2S2 M2S3 | 25.4 | 75 | 0.21 0.62 1.05 | W | 893 |
| M3S1 M3S2 M3S3 | 25.4 | 75 | 0.21 0.62 1.05 | W | 893 |
| M4S1 M4S2 M4S3 | 25.4 | 75 | 0.21 0.62 1.05 | W | 893 |

These types of conditions allow for steady-state simulations, which are well suited to evaluate different ventilation configurations, roof geometries, architectures, and dimensions of the structures in order to select the most appropriate design [40,41]. Therefore, these average conditions are valid for the object analysis in this study. Although it is recommended that future studies based on the conceptual basis and results of this study focus on the development of transient simulations on the final model selected, which should be evaluated under the hourly climatic conditions of the study region, which will allow the evaluation of a wider range of conditions of temperature, humidity, solar radiation, and wind speed and direction.

Once the numerical simulations were performed, we proceeded to the post-processing phase, where spatial curves are extracted from the qualitative behavior of the variables of interest such as speed and direction of air flows, temperature, and relative humidity inside each of the dryers. We also proceeded to extract the spatial numerical data of each of these variables for each of the scenarios evaluated, in this case, three planes (P1, P2, and P3) were drawn in the cross section of each dryer at a specific length from the facade of each dryer (Figure 4).

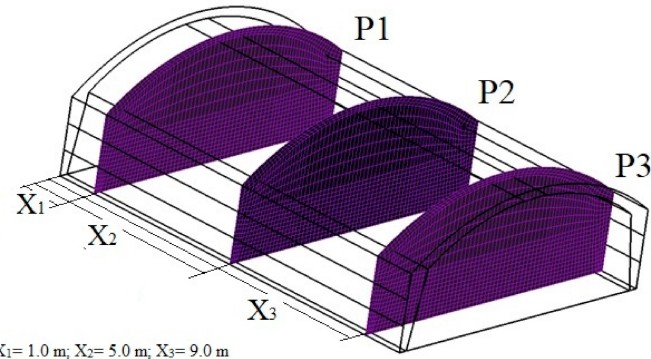

$X_1 = 1.0$ m; $X_2 = 5.0$ m; $X_3 = 9.0$ m

**Figure 4.** Geometric location of the analysis planes used for data extraction in the post-processing phase.

## 3. Results and Discussion

### 3.1. Effect of Dryer Design on Airflow Patterns

In the Figure 5, the characteristic velocities and patterns of the air flows inside the dryers evaluated under the established simulation conditions are observed. For the case of the M1 dryer, the air patterns show a similar behavior for each of the scenarios evaluated: S1, S2, and S3. For this case, it was found that the ventilation areas located on the east and west sides of the dryer prototype present a shared operation as simultaneous air inlet and outlet areas, something that usually happens due to the pressure changes that occur over these ventilation areas arranged on the side walls [42,43].

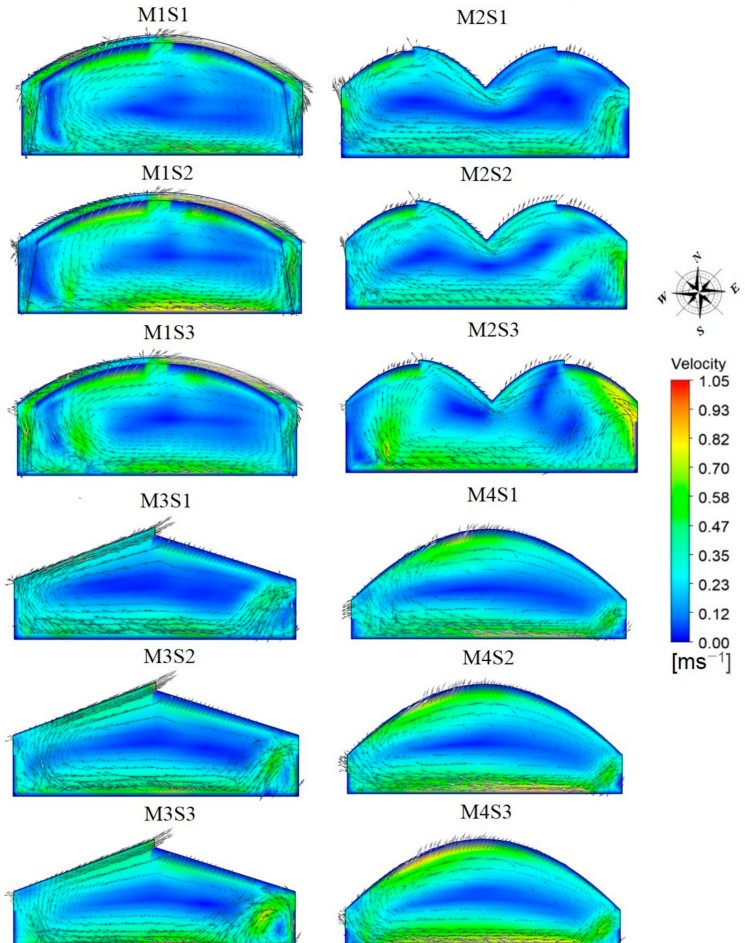

**Figure 5.** Spatial distribution and velocity of the simulated flow patterns (ms$^{-1}$), for each of the scenarios contemplated.

On the other hand, the largest air flows for M1 occur in the area near the ground and in the intermediate space contained between the inside and outside cover. The low velocity airflows occur in the central zone of the dryer, a zone that is influenced by a convective movement cell generated between the dryer roof and the soil region, this situation usually occurs in tunnel-type structures that do not have ventilation areas in the roof region [44].

The simulations also allowed to observe that the double plastic cover possibly allows to generate an acceleration of the air flows that move in the intermediate space existing between the two covers. This acceleration can be generated by the thermal differential between the polyethylene sheets of each roof and at the same time by the pressure differentials existing in this region, since the air flow inside a roof structure depends on the mentioned factors [45–48]. This is in agreement with what was reported in the study developed by Villagran et al. [45–48], who found that airflow patterns in naturally ventilated greenhouse-type structures are highly sensitive to changes in the roof of the structure being

evaluated. In the same way, Bournet el al. [45–48] reported that the airflows will depend on the specific location of the ventilation areas, which has been reaffirmed in different numerical studies of different types of roof structures that can be reviewed in the work developed by Bournet and Boulard [45–48].

In this same region, it can be seen that part of the airflow leaving the ventilation area arranged in the interior cover moves in two characteristic flows, one in an anti-clockwise direction that moves towards the side wall on the west side (W), following the shape of the dryer cover, and one with a similar type of movement, but in a clockwise direction, that moves towards the windward side wall (Figure 5). This airflow pattern observed in this study differs from the one reported in a structure with similar geometry but without the presence of a double cover in the work developed by Bartzanas et al. [49].

In the case of the M2 model, the airflow pattern generated is similar for scenarios S1 to S3 with the only differentiation being that as the outside wind speed increases the airflow velocity inside the dryer increases. In this case, it is observed that the airflow inlet to the dryer is through the side ventilation area of the windward wall, once the airflow enters the dryer it mixes with warm air and generates a downward flow that moves horizontally near the floor area to the leeward side wall, where part of the airflow from the dryer exits to the outside environment, this flow pattern is similar to that reported by Mistriotis et al. [50]. It can also be observed in M2 the generation of an airflow pattern that moves in the opposite direction of the outside wind, this displacement pattern follows the shape of the dryer roof and then leaves the interior of the structure through the ventilation areas located in the roof area.

In the case of M3, the windward wall shows that for the three speeds evaluated, once the air flow enters from the outside, it moves vertically to the soil region, and then horizontally to the leeward wall located on the west side (W) of the dryer and exits to the outside environment through the side and roof ventilation areas arranged in the structure. This flow pattern coincides with that reported by Perén et al. [51]. At the same time, it is observed that in the air inlet region of the dryer a movement loop is generated between the space contained from the side window to the floor, part of that air flow exits the dryer through the same air inlet area, a situation similar to that reported in a study developed with CFD by Akrami et al. [52].

Finally, for the M4 model, it is observed that the air flow patterns show a behavior where the air enters through the ventilation area arranged in the east side wall (E), once the air flow enters the dryer is accelerated in the area near the floor and then some of that air flow exits to the outside environment through the ventilation area of the leeward side wall (Figure 5). Likewise, it is observed that there is another characteristic air flow that moves following the shape of the roof in the opposite direction to the speed of the external wind, air flow that presents the same characteristics to those of a tunnel type structure evaluated in a natural ventilation scenario in a study developed by Couto et al. [53].

In numerical terms, in the post-processing phase approximately 2700 data were extracted in each of the planes (P1, P2, and P3) for each of the simulated scenarios, the calculated average air velocity values are shown in Table 5. For the M1 model, data were obtained ranging from a minimum of $0.299 \pm 0.184$ ms$^{-1}$ for P1 in the S1 simulation to a maximum of $0.349 \pm 0.167$ ms$^{-1}$ for P3 in the S3 simulation, therefore, it can be mentioned that as the outside wind speed increases from 0.21 (S1) ms$^{-1}$ to 1.05 ms$^{-1}$ (S3), the air velocity inside the dryer increases by an average of 16.7%.

**Table 5.** Temperature values obtained by simulation in each scenario contemplated.

| | Air Velocity (ms$^{-1}$) | | |
|---|---|---|---|
| Scenario | Plane 1 (P1) | Plane 2 (P2) | Plane 3 (P3) |
| M1S1 | 0.299 ± 0.184 | 0.308 ± 0.186 | 0.307 ± 0.185 |
| M1S2 | 0.319 ± 0.175 | 0.317 ± 0.173 | 0.314 ± 0.174 |
| M1S3 | 0.336 ± 0.172 | 0.329 ± 0.157 | 0.349 ± 0.167 |
| M2S1 | 0.269 ± 0.141 | 0.256 ± 0.137 | 0.259 ± 0.141 |
| M2S2 | 0.261 ± 0.136 | 0.271 ± 0.110 | 0.262 ± 0.152 |
| M2S3 | 0.302 ± 0.125 | 0.293 ± 0.152 | 0.297 ± 0.123 |
| M3S1 | 0.271 ± 0.156 | 0.275 ± 0.134 | 0.284 ± 0.128 |
| M3S2 | 0.295 ± 0.139 | 0.298 ± 0.141 | 0.288 ± 0.149 |
| M3S3 | 0.324 ± 0.123 | 0.313 ± 0.135 | 0.299 ± 0.136 |
| M4S1 | 0.336 ± 0.218 | 0.331 ± 0.210 | 0.325 ± 0.194 |
| M4S2 | 0.341 ± 0.214 | 0.340 ± 0.214 | 0.336 ± 0.206 |
| M4S3 | 0.346 ± 0.229 | 0.344 ± 0.227 | 0.333 ± 0.203 |

In the case of M2, the average velocity oscillated between a minimum value of $0.256 \pm 0.137$ ms$^{-1}$ at P2 for S1 and a maximum value of $0.299 \pm 0.125$ ms$^{-1}$ at P1 for S3, representing a 17.9% increase in velocity between scenarios. For M3, these minimum and maximum average velocities were $0.271 \pm 0.156$ ms$^{-1}$ and $0.295 \pm 0.139$ ms$^{-1}$ for P1 at S1 and S3, respectively, obtaining, therefore, an increase in velocity between scenarios of 8.8%. In M4, the highest average velocities were obtained, which ranged from a minimum and maximum value of $0.325 \pm 0.194$ ms$^{-1}$ to $0.346 \pm 0.229$ ms$^{-1}$ for P3 at S1 and for P1 at S3, so the velocity increase between these scenarios was only 6.4%. It should be mentioned that the air flows obtained for each of these dryers present higher velocities than those found in other types of dryers used at a regional level such as the one evaluated by Prada et al. [54].

In general terms, it can be observed that the average velocity variations between planes for the same simulation are minimal, which leads to the conclusion that the behavior of air flows is homogeneous inside each structure. Likewise, according to the characteristics of the air flows and the low wind speeds outside the region under study (< 1.1 ms$^{-1}$), it can be mentioned that the air movements are strongly related to the free convection phenomenon which causes a buoyancy effect by density changes in the fluid as a function of the temperature change [55,56].

### 3.2. Effect of Dryer Design on Spatial Temperature Distribution

The spatial temperature distribution for the three planes P1–P3 analyzed in each of the simulated scenarios is shown in Figure 6. In all the simulated dryer models, it was found that as the wind speed of the simulated outside environment increases, it is qualitatively observed that the temperature value decreases inside each of the dryers. Likewise, inside the structures the pattern of spatial distribution of temperature is maintained in each simulation S1, S2, and S3. This is due to the fact that the thermal distribution in structures that depend on natural ventilation are highly related to the speed and direction of the wind incident on the structure [40,41].

For the M2, M3, and M4 dryers, it was also found that the lowest temperature values were obtained, just in the area near the ventilation surface where the air flow from the outside enters. This flow of air as it moves through the interior of the dryer drags the heat towards the leeward wall where the highest values of temperature obtained in these three dryers can be observed, this situation is similar to the results found for greenhouse type solar dryers developed by Gupta et al. [57] and Purusothaman and Valarmathi [58]. In the case of M1, a differentiated behavior occurs since the regions with lower temperatures are located just above the one near the leeward and windward side walls, this may occur because in these places it was identified that the ventilation areas worked simultaneously as entry and exit air sites.

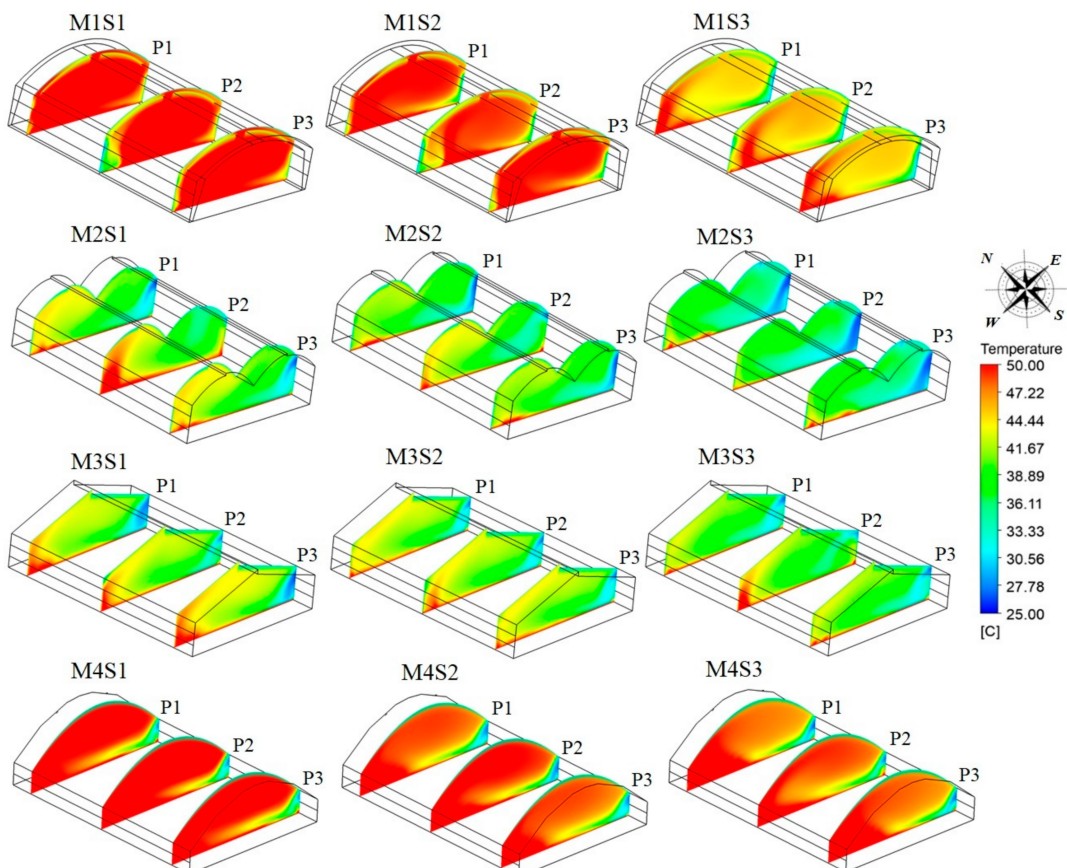

**Figure 6.** Spatial temperature distribution for all simulated scenarios.

Finally, it is also identifiable the increase of temperature that occurs inside each of the dryers. This is mainly due to the greenhouse effect produced by solar radiation and the low exchange of air that occurs between the dryer and the outside environment [59,60]. As solar radiation is the main source of energy supply, it can also be mentioned that the temperature differences observed between each of the dryers can be highly related to the geometric shapes of the roofs of the dryers evaluated [21].

In quantitative terms, the average temperature values obtained from the data extracted for each plane in each of the dryers and in all the simulated scenarios can be seen in Table 6. For the M1 model, the temperature varied between minimum and maximum average values of 43.38 ± 3.72 °C and 49.01 ± 1.31 °C for P1 on S3 and S1, respectively. For the M2 model, the same analysis shows maximum and minimum temperatures of 38.12 ± 3.75 °C and 41.10 ± 4.89 °C for P1 and P2 on S1 and S3, respectively.

For the M3 model the average temperature values ranged from a minimum of 38.49 ± 3.52 °C for P2 on S2 to a maximum of 41.30 ± 5.14 °C for P3. While for M4 these values were 45.52 ± 7.98 °C for the plane P3 in the simulation S3 and 46.87 ± 7.50 °C at P1 and S1, respectively. With these results, temperature reductions are obtained between scenarios S1 and S3 of 12.91%, 7.81%, 7.30%, and 2.96% in M1, M2, M3, and M4, respectively.

This temperature reduction is mainly influenced by the velocity and characteristics of the airflow patterns obtained for each dryer and more precisely by a higher rate of air renewal between the dryer and the outside environment as the wind speed incident on each structure increases [20,61]. It can also be mentioned that the most homogeneous thermal behavior for the same simulation scenario was obtained in the M1 model where the standard deviations of the extracted temperature data are the ones with the lowest value.

**Table 6.** Average temperature values obtained for each simulated scenario.

| | Temperature (°C) | | |
|---|---|---|---|
| Scenario | Plane 1 (P1) | Plane 2 (P2) | Plane 3 (P3) |
| M1S1 | 49.01 ± 1.31 | 48.43 ± 3.01 | 48.79 ± 2.98 |
| M1S2 | 47.14 ± 2.58 | 45.47 ± 2.71 | 47.18 ± 2.77 |
| M1S3 | 43.38 ± 3.12 | 43.49 ± 2.93 | 43.57 ± 2.96 |
| M2S1 | 39.25 ± 4.24 | 41.10 ± 4.89 | 39.28 ± 4.25 |
| M2S2 | 38.12 ± 3.75 | 39.22 ± 3.73 | 38.20 ± 3.80 |
| M2S3 | 35.83 ± 3.45 | 35.54 ± 3.38 | 35.97 ± 3.60 |
| M3S1 | 40.36 ± 4.03 | 40.36 ± 4.96 | 41.30 ± 5.14 |
| M3S2 | 40.04 ± 4.08 | 40.30 ± 3.91 | 40.06 ± 3.89 |
| M3S3 | 39.96 ± 4.05 | 38.49 ± 3.52 | 38.52 ± 3.57 |
| M4S1 | 46.87 ± 7.50 | 47.03 ± 7.66 | 46.38 ± 6.48 |
| M4S2 | 46.37 ± 7.17 | 46.15 ± 7.14 | 46.52 ± 7.67 |
| M4S3 | 45.96 ± 6.79 | 45.92 ± 6.75 | 45.52 ± 7.98 |

*3.3. Effect of Dryer Design on Spatial Distribution of Relative Humidity*

Another variable of interest that must be studied in the drying structures is the relative humidity. This variable has direct influence on the physical, chemical, and organoleptic conditions of the food products obtained at the end of a drying process [62]. Under conditions of not loading the product to be dried, the spatial distribution of the relative humidity must be related to the temperature distribution, therefore, regions with lower temperatures will be regions with higher relative humidity and vice versa [59].

In Figure 7, you can see the spatial distribution of relative humidity for each of the scenarios evaluated; in general terms, it was identified that the regions of higher relative humidity value coincide with the regions of wet air entry from the outside environment. Therefore, one of the recommendations for the operation of the dryer should focus on the management of ventilation areas during periods of high humidity, since inadequate management can affect the efficiency of the drying process [20].

The average relative humidity values obtained in each of the simulated scenarios can be found in Table 7. The relative humidity for the model M1 presented minimum and maximum values of 19.11 ± 3.61% and 27.34 ± 5.69% for the plane P1 in S1 and S3, respectively, in this model M1 was for which the lowest relative humidity values were obtained in the scenarios S1 and S2.

**Table 7.** Average relative humidity values obtained for each simulated scenario.

| Scenario | Relative Humidity (%) | | |
|---|---|---|---|
| | Plane 1 (P1) | Plane 2 (P2) | Plane 3 (P3) |
| M1S1 | 19.11 ± 3.61 | 20.33 ± 4.60 | 22.32 ± 7.31 |
| M1S2 | 21.93 ± 4.10 | 21.75 ± 4.21 | 24.50 ± 6.62 |
| M1S3 | 27.34 ± 5.69 | 26.81 ± 6.71 | 27.11 ± 6.41 |
| M2S1 | 34.70 ± 6.45 | 31.42 ± 7.34 | 34.93 ± 8.49 |
| M2S2 | 36.85 ± 8.32 | 34.52 ± 6.83 | 36.73 ± 8.32 |
| M2S3 | 42.33 ± 8.32 | 43.00 ± 8.96 | 42.16 ± 8.84 |
| M3S1 | 32.45 ± 7.34 | 31.62 ± 9.61 | 32.72 ± 6.63 |
| M3S2 | 33.21 ± 9.51 | 32.81 ± 7.53 | 33.18 ± 7.62 |
| M3S3 | 35.92 ± 7.32 | 36.93 ± 7.42 | 34.33 ± 6.91 |
| M4S1 | 22.21 ± 8.31 | 20.72 ± 9.42 | 22.02 ± 8.23 |
| M4S2 | 24.52 ± 10.3 | 23.31 ± 10.5 | 24.24 ± 10.2 |
| M4S3 | 24.92 ± 10.1 | 24.33 ± 9.42 | 25.03 ± 9.83 |

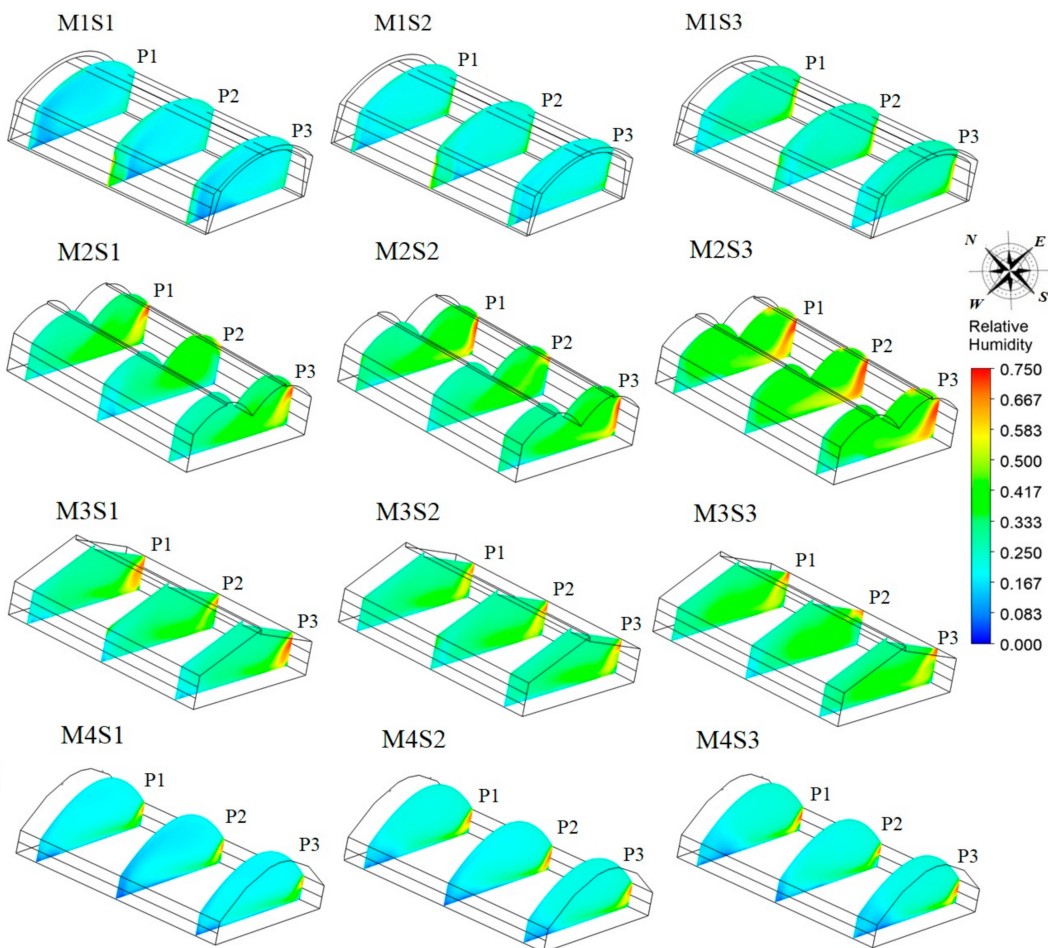

**Figure 7.** Spatial distribution of relative humidity for all simulated scenarios.

For the M2 dryer the relative humidity values ranged from a minimum of 34.70 ± 6.45% at P1 for S1 to a maximum value of 43.0 ± 8.96% in the P2 plane for the S3 simulation, which in turn was the highest humidity value obtained among all the simulations. In the case of the M3 dryer, the relative humidity values obtained were a minimum of 31.62 ± 9.61% and a maximum value of 36.93 ± 7.42% for the P2 plane at S1 and S3, respectively.

The average relative humidity values obtained in each of the simulated scenarios can be found in Table 7. The relative humidity for the model M1 presented minimum and maximum values of 19.11 ± 3.61% and 27.34 ± 5.69% for the plane P1 in S1 and S3, respectively, for which the lowest relative humidity values were obtained in the scenarios S1 and S2.

For the M2 dryer, the relative humidity values ranged from a minimum of 34.70 ± 6.45% at P1 for S1 to a maximum value of 43.0 ± 8.96% in the P2 plane for the S3 simulation, which in turn was the highest humidity value obtained among all the simulations. In the case of the M3 dryer, the relative humidity values obtained were a minimum of 31.62 ± 9.61% and a maximum value of 36.93 ± 7.42% for the P2 plane at S1 and S3, respectively.

In the M4 model, as it happened with the temperature variable, it was the model that presented the least variation in the value of relative humidity (Table 7). In this case, minimum and maximum values of relative humidity of 20.72 ± 9.42% and 25.03 ± 9.83% were obtained for the planes P2 and P3 in the scenarios S1 and S3, respectively. In general terms, it was found that for simulations without product loading the relative humidity value was less than 44%, values that are similar to those found by Prakash and Kumar [59] in an experimental study carried out in a solar dryer type greenhouse.

Finally, it can be mentioned that, of the prototypes evaluated, the models M1 and M4 are the ones that generate the highest temperature values with ranges between 43 and 49 °C,

although M1 presents a more homogeneous spatial distribution. The temperature values are within the optimal range recommended for drying mint (*Mentha spicata*), which is set between 40 and 50 °C, which ensures a high quality product for the infusion market [13,63]. On the other hand, Kripanand and Guruguntla [64] recommend natural convective drying at temperatures between 45 to 50 °C, as the best thermal range for achieving dehydrated products in this type of plant matrix. With these thermal conditions, it is possible for the product to be dried to reach a final moisture content close to 11% in an average time of 28 h as reported in the study developed by Nayak et al. [65], who evaluated the drying process of mint (*Mentha spicata*) in a greenhouse dryer with photovoltaic assistance with temperature conditions inside the dryer of 45.5 °C and a daily solar radiation of 520 W/m$^2$ in the external environment of the dryer.

Similarly, these temperature conditions were reported as optimal in the work developed by Nour-Eddine et al. [66], who also reported that temperatures below 50 °C allow preserving the essential oil fractions in Mentha species, which guarantees that this dry product can also be used in the industry therapeutic or cosmetic. It should be noted that the above reports agree with the results found in the simulation and modeling processes presented in this research. As for the relative humidity obtained for M1 and M4, these models presented values lower than 27.11% in all the simulated scenarios, relative humidity values that can be considered low with respect to the average value of the region of study. The above will undoubtedly help to improve the drying process of mint (*Mentha spicata*), for this case again a greater homogeneity is observed in the spatial distribution of relative humidity in the model M1.

Therefore, it could be said that the recommended model for the study region would be the M1 prototype dryer. This model should be evaluated in a future study under product loading conditions and under hourly scale climatic conditions, to determine the drying kinetics of the product. These future studies can take as a starting point the results of this research and thus complement the information through a field validated experiment under real operating conditions. In addition, this future study should consider different ways to place the vegetable material to be dried inside the prototype dryer to be built at full scale. It is also important to define the aerodynamic conditions of the mint (*Mentha spicata*) clusters such as porosity and drag coefficient in wind tunnel studies. Each of these physical, aerodynamic, and thermodynamic characteristics are necessary to establish the boundary conditions of a porous medium and thus be able to develop a realistic numerical simulation in a transient state, which will allow obtaining relevant results to optimize the mint (*Mentha spicata*) drying process.

## 4. Conclusions

The CFD model, previously validated, used in this research was efficient to determine the performance of the main variables that influence the process of solar drying of aromatic plants for four different types of dryer, without the need to perform the physical experiment, which could generate a high economic cost and obtain results in a longer period.

The air flows obtained inside each structure presented behavior patterns like those reported in other studies and where the dryer models were like those evaluated in this research, which allows giving greater validity to the CFD model used.

The tunnel type dryer with double plastic cover M1, is the model with the highest temperature values (49.01 ± 1.31) and lowest relative humidity values (19.11 ± 3.61) presented, which adjusts to the drying conditions required for aromatic plants in order to preserve their quality.

This M1 dryer has the highest microclimatic dynamics in response to the change in outside wind speed, therefore, the management of temperature and relative humidity through the ventilation areas can be more efficient for the purpose sought in the drying of aromatic plants.

The CFD modeling and simulation tool, due to its robustness to perform analysis in non-built scenarios, is an alternative for the development of experiments to contribute to the generation of efficient structure designs for the drying of aromatics plants.

**Author Contributions:** Conceptualization, E.V., J.C.H.-R., and G.F.; formal analysis, E.V.; investigation, E.V., J.C.H.-R., and G.F.; writing—original draft preparation, E.V. and J.C.H.-R.; writing—review and editing, E.V., J.C.H.-R., and G.F.; supervision, E.V., J.C.H.-R., and G.F. All authors have read and agreed to the published version of the manuscript.

**Funding:** This publication was financed by the Aurelio Llano Posada Foundation and the Corporación Colombiana de Investigación Agropecuaria (AGROSAVIA) and it did not receive external financing for the payment of APC.

**Institutional Review Board Statement:** Not applicable.

**Informed Consent Statement:** Not applicable.

**Data Availability Statement:** Not applicable.

**Acknowledgments:** The authors would like to thank the Corporación Colombiana de Investigación Agropecuaria (AGROSAVIA) for supporting this work. Work that was developed within the research project called Tecnificación del cultivo de Mentha spp en el Suroeste antioqueño con pequeños productores del programa Desarrollo Rural Integrado con Enfoque Territorial DRIET de la Fundación Aurelio Llano Posada. We also thank the farmers of Organización Campesinos Construyendo Futuro and the Aurelio Llano Posada foundation.

**Conflicts of Interest:** The authors declare no conflict of interest.

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
