# Peer review of "Thermo-Environmental Performance of Four Different Shapes of Solar Greenhouse Dryer with Free Convection Operating Principle and No Load on Product"

_fluids, doi:10.3390/fluids6050183_

Round 1

Reviewer 1 Report

Article Title:

Thermo-environmental performance of four different shapes of solar greenhouse dryer with free convection operating principle and no load on product

First of all, we appreciate all authors who contributed in this research work. To improve the article further it is good to clear the following points and incorporate at appropriate place.

  1. Page 2. Line 58. Clustering of literature is there [4,10,11]. Discuss each relevant article. Avoid clustering, and other places also.
  2. The model was evaluated using many average parameters (temperature, relative humidity, solar radiation…), making findings of performance highly tentative.
  3. What about precipitation and outside relative humidity and its influence?
  4. Line 300: Clustered citation to be extended with the typical results of [43], [44], [45]and [46].
  5. Discussion topic must be improved further. The drying conditions required for aromatic plants have not been adequately considered and discussed.
  6. Section 4. must be changed to 3.3.

Author Response

Dear
Reviewer, thank you very much for your kind words. Below you will find the answers to them. They have been a valuable input to enrich this new version of the article.

Page 2. Line 58. Clustering of literature is there [4,10,11]. Discuss each relevant article. Avoid clustering, and other places also.

Reply. We, the authors, have accepted this suggestion. We have therefore modified the text as follows between lines 56 to 61.

“While the active dryer needs the support of one or more fans to extract warm and humid air from inside the structure [4]. The use of these fans generates a higher degree of microclimatic management inside the dryer, which optimizes drying times and the drying process [10]. Additionally, this type of active dryers can be complemented with the use of renewable energies and heat accumulators to increase the internal temperature [11].”

Likewise on lines 75 to 80.

 “One of the most robust tools to simulate the microclimatic behavior of agricultural structures not built at full scale is the simulation from computational fluid dynamics  [15]. For example, in the study developed by Noh et al. [16] the authors studied the efficiency of three drying configurations in a new prototype industrial-type solar load dryer, finding through numerical simulation that the internal temperatures in the dryer could reach up to 59.8°C under an intermittent active ventilation configuration, which allowed optimizing the operation of the full-scale dryer”

The model was evaluated using many average parameters (temperature, relative humidity, solar radiation…), making findings of performance highly tentative. What about precipitation and outside relative humidity and its influence?

Reply. In accordance with the concern and in response to the reviewer and the readers of the manuscript, the authors have included the following text between lines 278 to 287.

“These types of conditions allow for steady-state simulations, which are well suited to evaluate different ventilation configurations, roof geometries, architectures and dimensions of the structures in order to select the most appropriate design. [55,56]. Therefore, these average conditions are valid for the object analysis in this study. Although it is recommended that future studies based on the conceptual basis and results of this study focus on the development of transient simulations on the final model selected, which should be evaluated under the hourly climatic conditions of the study region, which will allow the evaluation of a wider range of conditions of temperature, humidity, solar radiation and wind speed and direction.”

Line 300: Clustered citation to be extended with the typical results of [43], [44], [45]and [46].

Reply. We, the authors, have accepted this suggestion. We have therefore modified the text as follows between lines 315 to 323.

“This acceleration can be generated by the thermal differential between the polyethylene sheets of each roof and at the same time by the pressure differentials existing in this region, since the air flow inside a roof structure depends on the mentioned factors [43]. This is in agreement with what was reported in the study developed by Villagran et al. [44], who found that airflow patterns in naturally ventilated greenhouse-type structures are highly sensitive to changes in the roof of the structure being evaluated. In the same way Bournet el al. [45] They reported that the airflows will depend on the specific location of the ventilation areas, which has been reaffirmed in different numerical studies of different types of roof structures that can be reviewed in the work developed by Bournet and Boulard. [46].”

Discussion topic must be improved further. The drying conditions required for aromatic plants have not been adequately considered and discussed.

Reply.  The authors have included the following text in lines 484 to 500.

“The temperature values are within the optimal range recommended for drying mint (Mentha spicata), which is set between 40 and 50 °C, which ensures a high quality product for the infusion market [13,63]. On the other hand, Kripanand and Guruguntla [64] recommend natural convective drying at temperatures between 45 to 50 ⁰C, as the best thermal range for achieving dehydrated products in this type of plant matrix. These thermal conditions, it is possible for the product to be dried to reach a final moisture content close to 11% in an average time of 28 hours as reported in the study developed by Nayak, et al. [65], who evaluated the drying process of mint (Mentha spicata) in a greenhouse dryer with photovoltaic assistance with temperature conditions inside the dryer of 45.5⁰C and a daily solar radiation of 520W/m2 in the external environment of the dryer.

Similarly, these temperature conditions were reported as optimal in the work developed by Nour-Eddine, et al. [66], who also reported that temperatures below 50°C allow preserving the essential oil fractions in Mentha species, which guarantees that this dry product can also be used in the industry therapeutic or cosmetic. It should be noted that the above reports agree with the results found in the simulation and modeling processes presented in this research.”

Section 4. must be changed to 3.3.

Reply. The authors have included the suggestion.

Reviewer 2 Report

The manuscript “Thermo-environmental performance of four different shapes of solar greenhouse dryer with free convection operating principle and no load on product” is generally very well written and contains data of some relevance for a general readers as well as of high relevance for specialists in the topic. Computer simulations are very helpful and widely used in the design of new devices, machine elements and buildings. Although the subject of the paper could be of interest for the readers of the journal, the paper it should be supplemented with several analyzes.

  1. There is no information on how to arrange the material in the dryer, as filling the dryer with material will certainly change the air circulation and speed in the facility. It is a pity that the authors did not attempt to conduct such an analysis.
  2. There is no specific drying time for the material within the specified temperature range.
  3. When the temperature drops during drying (eg at night), the water may condense and the dried material may become wet. Is it possible to simulate the conditions in which this phenomenon occurs?
  4. There is no information on the strength and design aspects of these objects that could have an impact on the simulations (eg heat nodes).

Author Response

Dear
Reviewer, thank you very much for your kind words. Below you will find the answers to them. They have been a valuable input to enrich this new version of the article.

Reviewer # 2.

The manuscript “Thermo-environmental performance of four different shapes of solar greenhouse dryer with free convection operating principle and no load on product” is generally very well written and contains data of some relevance for a general reader as well as of high relevance for specialists in the topic. Computer simulations are very helpful and widely used in the design of new devices, machine elements and buildings. Although the subject of the paper could be of interest for the readers of the journal, the paper it should be supplemented with several analyzes.

  1. There is no information on how to arrange the material in the dryer, as filling the dryer with material will certainly change the air circulation and speed in the facility. It is a pity that the authors did not attempt to conduct such an analysis.

Reply. We agree with the reviewer that the loaded dryer will have different flow patterns, however the objective of this work is to select the best prototype to build. It is our goal to do this work in the future but for this we must determine the aerodynamic, thermodynamic, and physical characteristics of the material to be dried which will be done in a second phase of the project.

  1. There is no specific drying time for the material within the specified temperature range.

Reply. The authors have included the following text in the new version of the manuscript. lines 484 to 493.

“The temperature values are within the optimal range recommended for drying mint (Mentha spicata), which is set between 40 and 50 °C, which ensures a high quality product for the infusion market [13,63]. On the other hand, Kripanand and Guruguntla [64] recommend natural convective drying at temperatures between 45 to 50 ⁰C, as the best thermal range for achieving dehydrated products in this type of plant matrix. These thermal conditions, it is possible for the product to be dried to reach a final moisture content close to 11% in an average time of 28 hours as reported in the study developed by Nayak, et al. [65], who evaluated the drying process of mint (Mentha spicata) in a greenhouse dryer with photovoltaic assistance with temperature conditions inside the dryer of 45.5⁰C and a daily solar radiation of 520W/m2 in the external environment of the dryer.”

  1. When the temperature drops during drying (eg at night), the water may condense and the dried material may become wet. Is it possible to simulate the conditions in which this phenomenon occurs?.

Reply. The authors agree with the reviewer. This simulation should be done with the presence of the material to be dried and for the early morning hours which are periods where condensation can occur. The next phase is to build the full scale model and start the experimental evaluations that together with the activities of concern number 1 will give us the tools to perform a transient state simulation that will allow us to simulate the conditions under which this phenomenon could appear.

  1. There is no information on the strength and design aspects of these objects that could have an impact on the simulations (eg heat nodes).

           Reply. Finally, we would like to thank the reviewer for these suggestions and as authors we have included the following text at the end of the discussion section. Lines 506 to 519.

“Therefore, it could be said that the recommended model for the study region would be the M1 prototype dryer. This model should be evaluated in a future study under product loading conditions and under hourly scale climatic conditions, to determine the drying kinetics of the product. These future studies can take as a starting point the results of this research and thus complement the information through a field validated experiment under real operating conditions. In addition, this future study should consider different ways to place the vegetable material to be dried inside the prototype dryer to be built at full scale, it is also important to define the aerodynamic conditions of the mint (Mentha spicata) clusters such as porosity and drag coefficient in wind tunnel studies. Each of these physical, aerodynamic, and thermodynamic characteristics are necessary to establish the boundary conditions of a porous medium and thus be able to develop a realistic numerical simulation in a transient state, which will allow obtaining relevant results to optimize the mint (Mentha spicata) drying process.”

Grettings

The authors.

Round 2

Reviewer 1 Report

The manuscript has been significantly improved.